# High-Voltage Lithium-Ion Battery Using Substituted LiCoPO_4_: Electrochemical and Safety Performance of 1.2 Ah Pouch Cell

**DOI:** 10.3390/ma13194450

**Published:** 2020-10-07

**Authors:** Dongqiang Liu, Chisu Kim, Alexis Perea, Dubé Joël, Wen Zhu, Steve Collin-Martin, Amélie Forand, Martin Dontigny, Catherine Gagnon, Hendrix Demers, Samuel Delp, Jan Allen, Richard Jow, Karim Zaghib

**Affiliations:** 1Center of Excellence in Transportation Electrification and Energy Storage (CETEES), Hydro Québec, 1806 Boulevard Lionel-Boulet, Varennes, QC J3X 1S1, Canada; kim.chisu@hydroquebec.com (C.K.); Perea.alexis2@hydroquebec.com (A.P.); dube.joel@hydroquebec.com (D.J.); Zhu.wen@hydroquebec.com (W.Z.); Collin-Martin.steve2@hydroquebec.com (S.C.-M.); forand.amelie@hydroquebec.com (A.F.); dontigny.martin@hydroquebec.com (M.D.); gagnon.catherine3@hydroquebec.com (C.G.); Demers.Hendrix@hydroquebec.com (H.D.); samuel.delp.ctr@mail.mil (S.D.); jan.l.allen8.civ@mail.mil (J.A.); 2U.S. Army Research Laboratory, 2800 Powder Mill Road, Adelphi, MD 20783, USA; t.r.jow.civ@mail.mil; 3Department of Mining and Materials Engineering, McGill University, 845 Sherbrooke Street West, Montréal, QC H3A 0G4, Canada

**Keywords:** substituted LiCoPO_4_, ionic liquid, cycle life, safety, thermal stability

## Abstract

A LiCoPO_4_-based high-voltage lithium-ion battery was fabricated in the format of a 1.2 Ah pouch cell that exhibited a highly stable cycle life at a cut-off voltage of 4.9 V. The high-voltage stability was achieved using a Fe-Cr-Si multi-ion-substituted LiCoPO_4_ cathode and lithium bis(fluorosulfonyl)imide in 1-methyl-1-propylpyrrolidinium bis(fluorosulfony)imide as the electrolyte. Due to the improved electrochemical stability at high voltage, the cell exhibited a stable capacity retention of 91% after 290 cycles without any gas evolution related to electrolyte decomposition at high voltage. In addition to improved cycling stability, the nominal 5 V LiCoPO_4_ pouch cell also exhibited excellent safety performance during a nail penetration safety test compared with a state-of-the-art lithium ion battery. Meanwhile, the thermal stabilities of the 1.2 Ah pouch cell as well as the delithiated LiCoPO_4_ were also studied by accelerating rate calorimetry (ARC), thermogravimetric analysis (TGA), differential scanning calorimetry (DSC), and in situ X-ray diffraction (XRD) analyses and reported.

## 1. Introduction

Since its introduction by Amine et al. in 2000 [1], olivine-structured LiCoPO_4_ (LCP) has received particular attention as a promising cathode material for lithium-ion batteries owing to its high redox potential (4.8 V vs. Li^+^/Li) and theoretical capacity (167 mAh g^−1^). However, the poor electronic conductivity and low Li ion mobility in the lattice of LCP limits its practical application [2,3]. In the past decades, a variety of synthetic strategies, such as planetary milling, microwave heating, and spray pyrolysis have been explored to synthesize smaller particles and/or composites. However, LCP still has the above-mentioned shortcomings and exhibits an unsatisfactory electrochemical performance [4,5,6]. By depositing carbon coatings or precipitating Co_2_P under high-temperature annealing in an inert atmosphere, the electrical conductivity increased at least 10^5^ times compared to similar LiCoPO_4_ samples and undoped LiCoPO_4_ heated under air [2]. It was observed that the discharge capacity increased with increasing Co_2_P content to ~4–5 wt.%, after which the capacity rapidly decreased with increasing Co_2_P content [7]. Moreover, the attractiveness of LCP has improved by ion substitution, in both Li and Co sites [8]. Allen et al. reported that simultaneous substitution of Co in LCP by Fe, Cr, and Si ions resulted in 100% and over 80% capacity retentions over 250 cycles in a coin-type half-cell, and full cell, respectively [9]. In our previous work, we synthesized the same Fe-Cr-Si multi-ion-doped LCP and fabricated a preliminary 1.2 Ah pouch-type full cell [10]. Although, no capacity decay was observed for the coin-type half-cell over 120 cycles, the capacity of the 1.2 Ah pouch cell noticeably decreased after 20 cycles, which was mainly due to the decomposition of LiPF_6_-based organic carbonate electrolyte. Therefore, the limited electrochemical stability window of conventional electrolytes hinders the high-voltage cathode operation of LCP.

Ionic liquids (ILs) are non-flammable and have zero or an extremely low vapor pressure, properties, which are important factors for their application as safe electrolytes in lithium batteries [11]. So far, only a few ILs have been studied for LCP, and a pouch-type LCP full cell with ILs has not yet been reported. For instance, Xu et al. reported that, for a Li/Li_0.95_CoPO_4_ half-cell, better cycling stability and coulombic efficiency were achieved with 1 M lithium bis(trifluoromethanesulfonyl)imide in N-methyl-N-propyl pyrrolidinium bis(trifluoromethanesulfonyl)imide than with the conventional LiPF_6_-based organic electrolyte [12]. Lecce et al. [13] fabricated a Sn-C/LiFe_0.1_Co_0.9_PO_4_ coin-type full cell with lithium bis(tri-fluoromethanesulfonyl)imide (0.2 mol kg^−1^) in N-butyl-N-methylpyrrolidinium bis(trifluoromethanesulfonyl)imide, and obtained a limited capacity of 90 mAh g^−1^. Moreover, this coin-type full cell was cycled only for seven cycles.

In this study, we used 1.2 M lithium bis(fluorosulfonyl)imide (LiFSI) in 1-methyl-1-propylpyrrolidinium bis(fluorosulfonyl)imide (Py_13_FSI) as an electrolyte for our 1.2 Ah multi-stack pouch cells comprising substituted LCP as the cathode and natural graphite as the anode. We evaluated the high voltage stability and rate performance of the pouch cell at 25 °C. In addition, we performed nail penetration and hot box tests to evaluate the safety performance of the pouch cell under electrical abuse conditions and determined the thermal stability of delithiated LCP by accelerating rate calorimetry (ARC), thermogravimetric analysis (TGA), differential scanning calorimetry (DSC), and in situ X-ray diffraction (XRD) analyses.

## 2. Experimental

### 2.1. Full Cell Fabrication and Electrochemical Tests

The Fe-Cr-Si-doped LCP cathode materials were synthesized by high-energy ball milling [10]. The cathode was prepared by mixing LCP, Super P, carbon nanotube and polyvinylidene difluoride (PVDF) with the weight ratio of 90:3.5:1.5:5 in N-methylpyrrolidine (NMP) and coated on aluminum foil. The anode was composed of nature graphite, vapor grown carbon fibers and PVDF in the weight ratio of 94:1:5 and dispersed in NMP under continuous stirring and coated on copper foil. The electrodes were then pressed and punched into 49 mm × 52 mm and 51 mm × 55 mm for cathode and anode, respectively, before drying under vacuum. The graphite/LCP pouch-type full cells with a nominal capacity of 1.2 Ah were fabricated with 19 stacks of LCP and 20 stacks of graphite. Instead of an ethylene carbonate (EC) and ethyl methyl carbonate (EMC)-based electrolyte as described in ref. [10], we used a room-temperature IL, Py_13_FSI, (Dai-Ichi Kogyo, Seiyaku Co., Ltd., Kagawa, Japan), and 1.2 M LiFSI, (Nippon Shokubai, Osaka, Japan) as the electrolyte. The IL contains less than 10 ppm (w/w) H_2_O and less than 2 ppm (w/w) halide and alkali metal-ion impurities. The electrochemical measurements were galvanostatically performed using a battery cycler (BCS815, BioLogic, Seyssinet-Pariset, France) at 25 °C in the voltage range of 3.5–4.9 V. The assembled pouch cell was first cycled at 0.1 C rate (1C = 1.19 A) for 2 cycles and then cycled at 0.2 C and 25 °C. After every 20 cycles, electrochemical impedance spectroscopy (EIS) was performed at the end of discharge in a potentiostatic mode in the frequency range 10 kHz to 50 mHz with an amplitude of 5 mV.

### 2.2. Safety Tests

After cycling, nail penetration and hot box tests were performed on the 1.2 Ah pouch cells using in-house equipment. Before the safety tests, the experimental cells were fully charged to 4.9 V at 0.1 C rate and potentiostatically maintained at 4.9 V for 24 h. The nail penetration test was performed by penetrating a 3-mm-diameter stainless steel rod at a rate of 0.5 cm s^−1^. The hot box test was carried out in an oven in the temperature range of 20–170 °C at a heating rate of 5 °C min^−1^. The cell voltage and cell surface temperature were monitored using a data logger during the safety tests. The temperature was measured using K-type thermocouples attached on the cell surface in the center and at the four corners of the cell.

### 2.3. Accelerating Rate Calorimetry

The thermal runaway characteristics of the pouch cell at 100% state-of-charge (SOC) were investigated by ARC using an accelerating rate calorimeter (Thermal Hazard Technology, Milton Keynes, UK). The ARC measurements were conducted with a heating step of 5 °C in the temperature range of 50–400 °C. The maximum exothermal rate was 20 °C min^−1^. Prior to measurements, the thermocouples of the calorimeter were calibrated using a steel plate of the same size as that of the pouch cell. The diameter and depth of the calorimeter were 10 cm. In a typical ARC test, the pouch cell is placed in an adiabatic calorimeter, heated to 50 °C, and kept at this temperature for 40 min (rest step) to stabilize the cell temperature. During the rest step, the calorimeter records the self-heating process with a sensitivity threshold of 0.02 °C min^−1^. If no exothermic reaction is detected, the calorimeter is heated with a 5 °C step and the sequence is repeated. If an exothermic reaction is detected that is greater than the sensitivity threshold, the calorimeter switches to the adiabatic mode to follow the changes in the cell surface temperature.

### 2.4. Thermogravimetric Analysis and Differential Scanning Calorimetry

To study the thermal behavior of the delithiated Fe-Cr-Si-doped LCP cathode material, a graphite/LCP pouch cell was fully charged to 4.9 V in the same constant current and constant voltage mode as mentioned for ARC analysis. The cell was then disassembled in a glove box and the LCP positive electrode was retrieved, rinsed with dimethyl carbonate (DMC) three times, and dried at 100 °C under vacuum for 2 h. Then, approximately 20 mg of LCP sample was carefully scratched and collected from the positive electrode for TGA and DSC measurements. TGA was performed in the temperature range of 30–700 °C at a heating rate of 10 °C min^−1^ under a constant airflow rate of 90 mL min^−1^ using a TA Instruments TGA Q500/Discovery MS thermal analyzer (New Castle, DE, USA). DSC was performed in the temperature range of 50–170 °C at a heating rate 10 °C min^−1^ under nitrogen flow using a Perkin Elmer Pyris 1 differential scanning calorimeter (PerkinElmer, Boston, MA, USA).

### 2.5. In Situ X-ray Diffraction Analysis

To determine the thermal stability of the delithiated LCP powders, in situ XRD analysis was carried out using a Smartlab diffractometer (Rigaku, Tokyo, Japan) with Co-Kα radiation. The delithiated LCP samples collected from a 100% SOC pouch cell and those analyzed by TGA-DSC were further examined by XRD analysis by heating from 20 °C to 400 °C at a heating rate of 5 °C min^−1^ under nitrogen flow (30 mL min^−1^). The temperature was maintained at 400 °C for 1 h and then decreased to 34 °C and the data were collected every 5 min with a scan step size of 0.04° in the 2θ range of 10°–60°.

## 3. Results and Discussion

### 3.1. Electrochemical Performance of 1.2 Ah LCP Pouch Cell

In our previous work [10], the same 1.2 Ah pouch cells were fabricated with 1.2 M LiPF_6_ in EC-EMC as the electrolyte. These cells exhibited an initial capacity of 120 mAh g^−1^, which rapidly decreased to 95 mAh g^−1^ after 20 cycles even under an appropriate pressure with considerable gas evolution (Appendix A), mainly due to the decomposition of the carbonate-based electrolyte at a high charging voltage. ILs are considered good alternatives to conventional electrolytes for high-voltage cathodes because of their excellent stability at high voltage (>4.0 V) and outstanding safety owing to their non-flammability [14]. A drawback of ILs is their high viscosity, which leads to a low ionic conductivity and low power performance. In this study, LiFSI and Py_13_FSI were used as the electrolyte for the high-voltage LCP cathode owing to their high stability under high voltage conditions and higher ionic conductivity than other types of ILs. To the best of our knowledge, this is the first study in which LiFSI and Py_13_FSI are used as the electrolyte for a high-voltage LCP cathode. Unlike in the case of the carbonate-based electrolyte, gas evolution did not occur during cycling for the pouch cell comprising the new Py_13_FSI IL-based electrolyte in this study (Appendix A). This suggests that the Py_13_FSI-based electrolyte did not decompose because of its wide chemical stability window [14]. Figure 1a shows the voltage profiles of the 1.2 Ah graphite/LiFSI-Py_13_FSI/LCP pouch cell cycled at 0.2 C rate in the voltage range of 3.5–4.9 V. As can be seen, a sloping region centered around 4.65 V followed by a flat plateau at ~4.75 V appeared during charging, while a sloping voltage profile was obtained upon discharge; this indicates the presence of a solid-solution and the two-phase behavior of the Co–Fe–Si-substituted LiCoPO_4_ cathode [15]. In addition, a reversible capacity of 1.19 Ah (specific capacity 120 mAh g^−1^) was obtained with the LiFSI-Py_13_FSI electrolyte, which implies that the Py_13_FSI–based electrolyte can be used not only for a 3.5 V olivine LiFePO_4_ (LFP) cathode [16], but also for a 5 V olivine LCP cathode. Moreover, the slight increase in charge voltage and decrease in discharge voltage with cycling indicate a slight increase in cell polarization during cycling. Figure 1b shows the cycle life and capacity retention of the pouch cell at 0.2 C rate and 25 °C. The pouch cell exhibited an initial capacity of 1.19 Ah (specific capacity 120 mAh g^−1^) and retained a capacity of 1.06 Ah (specific capacity 107 mAh g^−1^) after 290 cycles with 91% of capacity retention. Table 1 shows the performances of the 5 V LCP pouch full cell fabricated in this study and other LCP-based full cells reported in literature. To the best of our knowledge, the graphite/LiFSI-Py_13_FSI/LCP pouch cell fabricated in this work exhibits the best performance with a long cycle life and higher capacity retention than those of any other LCP-based full cells reported in the literature. The superior cycling performance of the graphite/LiFSI-Py_13_FSI/LCP pouch cell is attributed to the solid-solution domains in LCP that reduced the mechanical strain during electrode operation [17] and the exceptional stability of the Py_13_FSI electrolyte at a high voltage. Moreover, although the durability was improved, the 1.2 Ah graphite/LiFSI-Py_13_FSI/LCP pouch cell exhibited approximately 9% capacity loss compared with the 100% capacity retention of the Fe-Cr-Si multi-ion-substituted LCP coin-type half-cell [9,10]. Since the electrolyte did not decompose during cycling, the capacity loss of the full cell can be attributed to the growth of a solid electrolyte interface (SEI) with cycling on both the cathode and anode [9,18,19,20,21], which irreversibly consumed Li, thereby increasing the cell polarization.

EIS was performed to better understand the electrochemical performance. The Nyquist plots of the 1.2 Ah pouch cell after every 20 cycles at the end of discharge are shown in Figure 1c. As can be seen, all the Nyquist plots display a semicircle in the high-frequency region. The semicircle intercept with the real axis represents the ohmic resistance of the cell, which can be attributed to the interfacial electrolyte resistance, electronic resistance of the electrode particles and current collectors, and the connection resistance between the cell terminals and instrument leads [21]. As shown in Figure 1c, the ohmic resistance of the cell increased with cycling, which can be attributed to the formation and reformation of the Li-consuming SEI layer on the electrodes and the increase in interfacial resistance. This also explains the increase in cell polarization (Figure 1a) and the capacity decay (Figure 1b). Furthermore, the slope of the straight line in the low-frequency region of the Nyquist plots (Figure 1c) represents the Warburg resistance, which is related to the diffusion of Li^+^ ions in the bulk particles. Because of the porous electrode, the effect of electrolyte, and the capacitive blocking behavior, the angle at which the straight line is inclined to the real axis in the low-frequency region is greater or smaller than 45° [21]. As can be seen in Figure 1c, the slope of the line did not change with cycling, which indicates that the electrolyte did not decompose during the test.

Figure 1d presents the rate capability of the 1.2 Ah graphite/LiFSI-Py_13_FSI/LCP pouch cell at 25 °C. The discharge capacity remained almost the same (1.2 Ah) from 0.2 C to 0.5 C with only a slight reduction of less than 1%. The capacity decreased to approximately 1 Ah at 1 C rate and less than 0.5 Ah at 2 C rate. It should be noted that the Fe-Cr-Si-substituted LCP cathode exhibited an enhanced rate performance in a coin-type half-cell using commercial carbonate electrolyte [9,10]. However, the viscosity of Py_13_FSI (39 mPa s) is over five times higher than that of EC-DEC (7.68 mPa s) and the olivine LiFePO_4_ cathode exhibits limited power performance [16]. Therefore, the lower rate capability of the 1.2 Ah graphite/LiFSI-Py_13_FSI/LCP pouch cell can be ascribed to the high viscosity of the IL electrolyte.

### 3.2. Safety Studies of 1.2 Ah LCP Pouch Cell

To evaluate the safety of the 5 V graphite/LiFSI-Py_13_FSI/LCP pouch cell, a nail penetration test was performed at the end of charge (100% SOC). Prior to the safety test, the experimental cells were fully charged to 4.9 V at 0.1 C rate and potentiostatically maintained at 4.9 V for 24 h. For comparison, the nail penetration test was also performed on an iPhone 6 Plus cell, in which the cell was galvanostatically charged to 4.4 V instead of 4.9 V before testing. Figure 2a,b show the photograph, temperature and voltage change of the LCP and iPhone pouch cells before, during, and after the nail penetration test, respectively. Appendix A show the nail penetration testing of the samples. As shown in Figure 2a, the LCP pouch cell exhibited a temperature of 24.7 °C with an open circle voltage (OCV) of 4.754 V before testing. Neither smoking or sparking, nor cell short-circuiting occurred with nail penetration; only the OCV slightly decreased and the temperature gradually increased. After 10 min, the cell temperature increased to a maximum of 52.2 °C, and the OCV gradually decreased to 4.510 V. After 30 min, both the OCV and cell temperature continuously decreased until the end of the test. The graphite/LiFSI-Py_13_FSI/LCP pouch cell maintained its structure even after the nail penetration test, with only a hole in the center of the pouch cell. In contrast, the nail penetration test performed on the iPhone cell led to drastic consequences (Figure 2b): Immediately after the nail was inserted, the cell shorted and smoke started to emanate from the cell. Then sparks flashed and the cell burned with a large amount of smoke, and the iPhone cell temperature increased to ~358 °C in 5 s. The entire cell was completely ruined after the test. The nail penetration test demonstrated the excellent safety performance of the 1.2 Ah graphite/LiFSI-Py_13_FSI/LCP pouch cell compared with the commercial iPhone cell.

A hot box test was also carried out to determine the safety performance of the pouch cell under thermal abuse conditions. Before the test, the cell performance was confirmed under different conditions. The cell was then charged to 100% SOC, placed in an oven, and heated from 20 °C to 170 °C at a heating rate of 5 °C min^−1^. Appendix A shows the photograph of the pouch cell before heating. Five thermocouples (T.C. 1–5) were attached on the cell surface to measure the temperature change, and another thermocouple (T.C. 6) was used to detect the oven temperature. The changes in cell temperature and voltage with time are presented in Figure 3, where T.C. 1–6 represent the temperature measured by the six thermocouples and the green line at the top represents the cell voltage. As can be seen from Figure 3, with an increase in temperature from 20 °C to ~150 °C, the cell voltage remained unchanged and the OCV remained constant at 4.73 V. However, with a further increase in temperature to over 150 °C, the cell temperature surpassed the oven temperature, which indicates the initiation of exothermic reactions inside the cell, the cell voltage rapidly decreased, and the OCV suddenly dropped to 0 V. The cell temperature quickly reached the maximum of ~280 °C. After the hot box test, the cell swelled and burst, as shown in Appendix A, which indicates that the exothermic decomposition reactions were triggered at ~150 °C. Since the IL Py_13_FSI is stable below 200 °C [22], it is reasonable that the exothermic decomposition reactions are related to the LCP cathode at 100% SOC.

The thermal runaway behavior of the 1.2 Ah graphite/LiFSI-Py_13_FSI/LCP pouch cell at 100% SOC was further investigated by ARC. For this, 2 Ah pouch cells employing the same anode but different cathodes (LiCoO_2_ (LCO) and Li[Ni_0.6_Co_0.2_Mn_0.2_]O_2_ (NCM)), and a 3 Ah LFP cylindrical cell were compared. Before the test, the experimental cells were fully charged and cut off at 4.4 V for the LCO and NCM cells and at 3.65 V for the LFP cell. The preliminary test data obtained from the ARC heat-wait-search test are presented in Figure 4a and the corresponding self-heating rate (°C min^−1^) vs. temperature plots are presented in Figure 4b. The onset, thermal runaway, and maximum temperatures are summarized in Table 2. The onset temperature is reached when the self-heating rate is greater than the detection limit of the accelerating rate calorimeter (0.02 °C min^−1^). The first values of the self-heating rate associated with cell degradation are probably related to the start of SEI degradation [23]. The SEI film is composed of stable and metastable components such as Li_2_CO_3_ and (CH_2_OCO_2_Li)_2_, depending on the composition of the carbonate-based electrolyte [24]. The onset temperature of the LCP cell (77 °C) is close to that of the LCO cell (82 °C) but lower than those of the NCM (117 °C) and LFP (106 °C) cells. Since there is no direct correlation observed between the onset temperature and cell size [23], the onset temperatures in Table 2 indicate that the composition of the SEI in the LCP cell employing the LiFSI-Py_13_FSI electrolyte is different from the composition of the SEI in the LCO, NCM and LFP cells employing the carbonate-based electrolyte and more sensitive with the temperature.

This is a repeated paragraph as in page 8. After SEI degradation, the electrodes are no longer protected from contact with the electrolyte, thus, the exothermic reaction continues due to the direct reaction of the electrode material with the electrolyte components. In the temperature range of 120–150 °C, no exothermic reaction was detected in all the batteries, which is probably because of the melting of the binder, separator, and/or cell venting and energy absorption due to electrolyte evaporation. The self-heating rate continued to increase even after the separator melted, and the thermal runaway was dominated by the cathode side reactions. As shown in Table 2, among the tested cells, the LCP cell displayed the lowest thermal runaway temperature of 168 °C. Moreover, the maximum temperature of the LCP cell (330 °C) was also the lowest, which might be related to the smaller size of the 1.2 Ah compared with the other cells. Note that the oxygen from the transition metals in the cathode material reacts with the organic-based electrolyte, which readily combusts when the temperature increases to approximately 180 °C [23]. Since the IL Py_13_FSI is stable below 200 °C [22], the lowest thermal runaway temperature of 168 °C of the LCP cell is attributed to the delithiated LCP cathode at 100% SOC, which is consistent with the short-circuiting at 150–170 °C during the hot box test.

### 3.3. Thermal Stability Studies of Substituted LCP Cathode Material

To further evaluate the thermal stability of the delithiated cathode material, LCP samples were collected for TGA and DSC analyses from the same 1 Ah LCP pouch cell at 100% SOC that was used for ARC measurements. Figure 5a displays the TGA curves of the delithiated and fresh (lithiated) LCP powders, the corresponding DSC profiles are presented in Figure 5b. As shown in Figure 5a, no mass loss can be observed for the fresh LCP powder up to 550 °C. In contrast, the delithiated LCP exhibited approximately 2.5% weight loss in the temperature range of 100 °C to 131 °C, with a corresponding exothermic peak at ~115 °C in the DSC profile (Figure 5b). In the delithiated LCP samples, the residual PVdF-based binder is expected to decompose at temperatures higher than 350 °C [25]; hence, the weight loss and exothermic reaction are mainly due to the decomposition of delithiated LCP. Bramnik et al. [26] also reported that olivine Li*_x_*CoPO_4_ decomposes in the range 100–200 °C and Li_0.2_CoPO_4_ decomposes faster than Li_0.7_CoPO_4_. The exothermic reaction accelerated at 130 °C (Figure 5b) and additional weight losses were observed at 196 °C and 305 °C (Figure 5a). The decomposition of delithiated LCP accompanied by gas evolution and pressure build-up, led to cell rupture during the hot box test and ARC measurements.

To better understand the thermal stability of delithiated LCP, in situ XRD analysis was performed under nitrogen flow in the temperature range of 20–400 °C, and the results are shown in Figure 6. As can be seen, a small amount of partially delithiated Li_0.7_CoPO_4_ and traces of Li_9_Cr_3_P_2_O_7_ impurities are present at 100% SOC at 20 °C. With an increase in temperature to ~108 °C, the peak intensity of the CoPO_4_ phase started to decrease, which indicates that the CoPO_4_ phase started to decompose. This is consistent with the appearance of an exothermic peak at ~115 °C in Figure 5b. However, the Li_0.7_CoPO_4_ phase remained unchanged at 108 °C. Meanwhile, an unknown phase at 2θ = 21° appeared that quickly disappeared with subsequent reactions. Further, traces of Cr_2_O_3_ arising from the CoPO_4_ phase was observed at ~120 °C, and the peak intensities of the Li_0.7_CoPO_4_ phase started to decrease and almost no Li_0.7_CoPO_4_ phase was detected above 163 °C. Above 200 °C, Co_3_(PO_4_)_2_ and Co_2_P_2_O_7_ phases were mainly detected along with a trace amount of Li_9_Cr_3_P_2_O_7_ impurity until the end of the test. It is noteworthy no decomposition product of the Li_9_Cr_3_P_2_O_7_ phase was detected during the in situ XRD measurements. As can be seen from Figure 6, the decomposition reaction of the olivine Li*_x_*CoPO_4_ phase started at ~108 °C and ended at ~200 °C. The decomposition of carbon-coated CoPO_4_ proceeds with the evolution of O_2_ and CO_2_ gases. Therefore, the overall decomposition reaction can be written as:5CoPO_4_ + C = Co_3_(PO_4_)_2_ + Co_2_P_2_O_7_ + *½* P_2_O_5_ + CO_2_ ↑ + *¼* O_2_ ↑(1)

As shown in reaction (1), the unknown phase (pink circle at 2θ = 21°) in Figure 6 can be ascribed to P_2_O_5_ (JCPDS 023-1301). Furthermore, the calculated weight loss in Reaction (1) is 6.5%, which is close to the weight loss of 5% at 196 °C observed in Figure 5a. The difference in weight loss is probably because of the Fe–Cr–Si substitutions in the LCP phase. Bramnik et al. reported the formation of only the Co_2_P_2_O_7_ phase after Li*_x_*CoPO_4_ decomposition [26], while in our study, Co_3_(PO_4_)_2_ was the dominant phase with a small amount of Co_2_P_2_O_7_ phase (Figure 6); this indicates the different decomposition mechanisms of the unsubstituted LCP [26] and Fe–Cr–Si-substituted LCP. Although, Fe–Cr–Si-substitution enhanced the cycling performance (Figure 1), the low thermal stability of delithiated Li*_x_*CoPO_4_ still needs to be improved. It is generally believed that highly covalent P-O bonds (596 kJ mol^−1^) prevent the evolution of oxygen from olivine-structured FePO_4_. However, in situ XRD analysis revealed that the thermal stability of isostructural olivine-like CoPO_4_ significantly depends on the 3*d* metal. In an octahedral field, Fe^3+^ has a 3*d* electronic configuration of $t_{2g}^{3}e_{g}^{2}$, which is more stable than the 3*d* electronic configuration of Co^3+^ ($t_{2g}^{4}e_{g}^{2})$. Moreover, the lower energy of the Co–O bond (368 kJ mol^−1^) than that of the Fe–O bond (409 kJ mol^−1^) [27] may also explain the intrinsic low thermal stability of CoPO_4_.

## 4. Conclusions

We fabricated 1.2 Ah pouch cells with a natural graphite anode and Fe–Cr–Si multi-ion-substituted LCP cathode using an IL-based (LiFSI in Py_13_FSI) electrolyte. Unlike the carbonate electrolyte-based cell, the IL-based cells did not exhibit gas evolution during prolonged cycling. Furthermore, the LCP cell delivered an initial capacity of 1.19 Ah at 0.2 C rate and maintained a capacity of 1.07 Ah after 290 cycles. The safety performance of the 1.2 Ah pouch cell at 100% SOC was investigated by nail penetration, and hot box test and ARC and compared with that of an iPhone cell. The iPhone cell immediately shorted and burned during nail penetration test. In contrast, the 5 V LCP pouch cell with the Py_13_FSI electrolyte exhibited an excellent safety performance with no trace of even smoke or sparks during the electrical abuse test. However, ARC, TGA-DSC, and in situ XRD analyses suggest that delithiated LCP is thermally unstable at high temperature above 115 °C. Therefore, further work is necessary to improve the thermal stability of LCP to realize the commercialization of LCP-based batteries.

## Figures and Tables

**Figure 1 materials-13-04450-f001:**
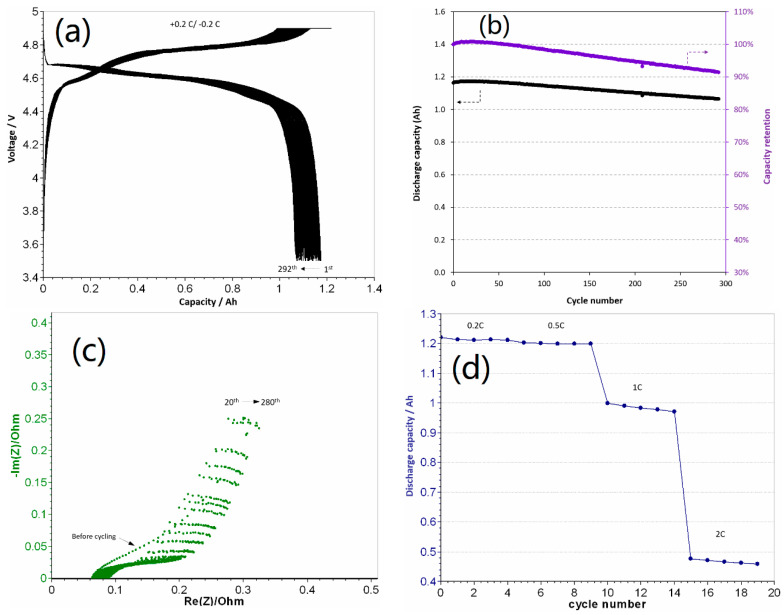
Voltage profiles of the 1.2 Ah graphite/LiFSI-Py_13_FSI/LCP pouch cell under 0.2 C rate (**a**); cyclability of the 1.2 Ah LCP pouch cell under 0.2 C rate (**b**); Nyquist plots of the 1.2 Ah LCP pouch cell every 20 cycles at the end of discharge during cycling (**c**) and rate performance of the 1.2 Ah LCP pouch cell at 25 °C (**d**).

**Figure 2 materials-13-04450-f002:**
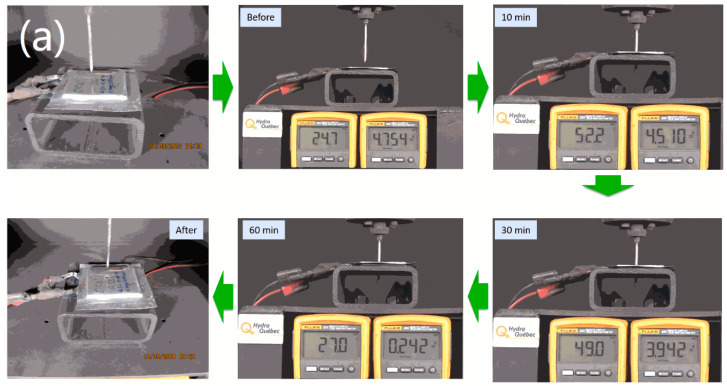
A Photographs, temperature and voltage change of the 1.2 Ah LCP (**a**) and iPhone (**b**) pouch cells before, during, and after the nail penetration test.

**Figure 3 materials-13-04450-f003:**
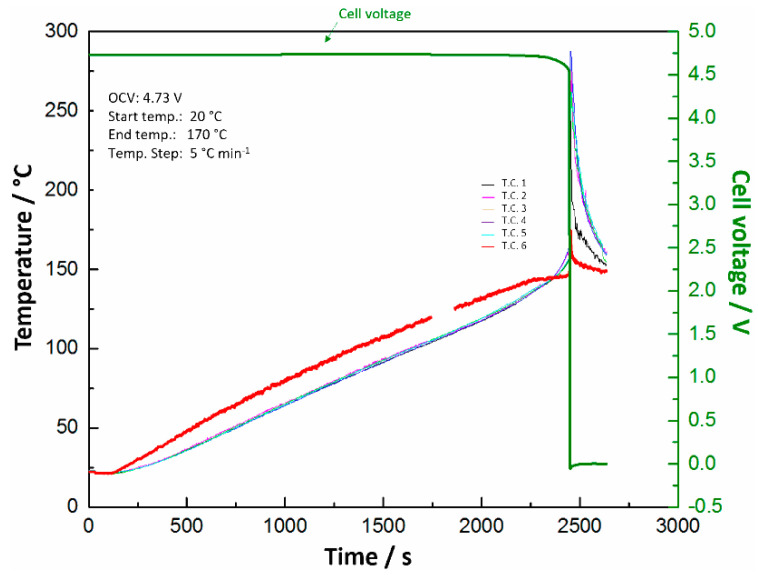
Evolution of the cell temperature and voltage with time during hot box test.

**Figure 4 materials-13-04450-f004:**
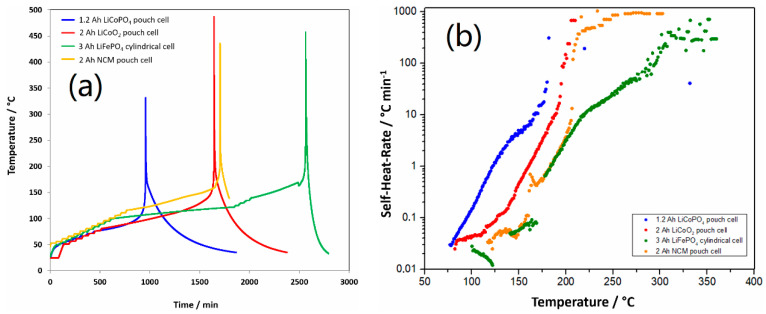
Temperature versus time for thermal decomposition of selected batteries at 100% SOC (**a**) and the dependence of self-heating-rate on temperature of selected batteries at 100% SOC (**b**). The thermal runaway temperature is defined as the temperature self-heat-rate ≥10 °C min^−1^.

**Figure 5 materials-13-04450-f005:**
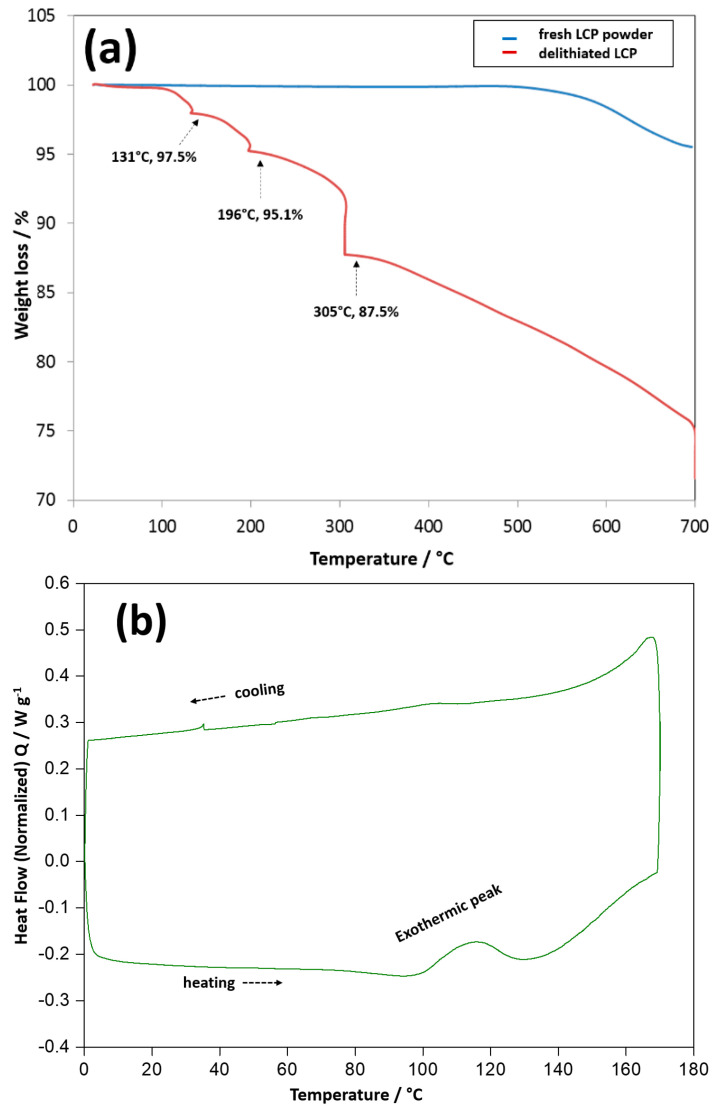
TGA curves of the fresh (lithiated) and delithiated LCP samples (**a**) and DSC curve of delithiated LCP powders (**b**).

**Figure 6 materials-13-04450-f006:**
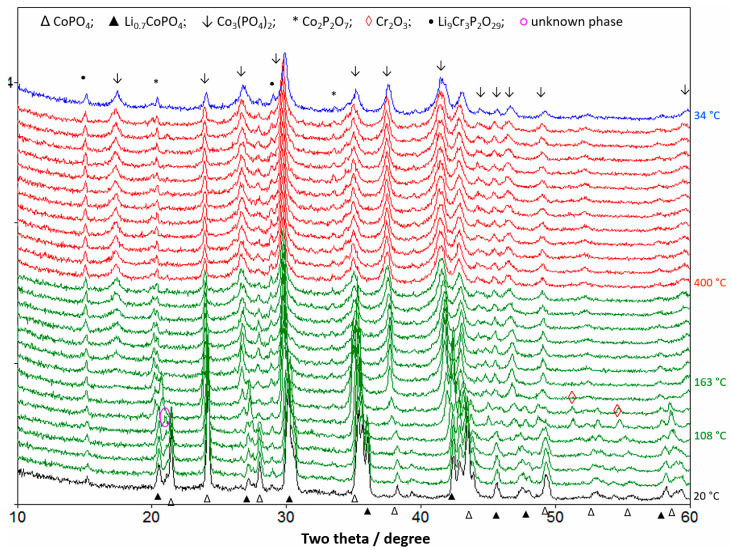
In situ XRD patterns of the delithiated LCP under N_2_ upon heating from 20 to 400 °C and subsequent cooling from 400 to 34 °C at a heating rate of 5 °C min^−1^.

**Table 1 materials-13-04450-t001:** Comparison of performances of the 5 V LiCoPO_4_ pouch cell and the full cells reported in literature.

	Sn-C/Doped-LCP	Gr/Doped-LCP	LTO/LCP	Gr/Doped-LCP
Electrolyte	0.2 mol Kg^−1^ LiTFSI/Pyr12TFSI	1.2 M LiPF_6_/EC-EMC	1.0 M LiPF_6_/EC-DMC-DEC	1.2 M LiFSI/Py_13_FSI
Cell type	Coin cell	Coin cell	Coin cell	Pouch cell
C-rate	0.05	0.33	0.1	0.2
Specific capacity (mAh g^−1^)	90	120	144	120
Cycle number	7	250	100	290
Capacity retention (%)	83	>80	93.1	91
Ref	13	9	18	This work

**Table 2 materials-13-04450-t002:** Onset, thermal runaway, and maximum temperatures of the examined batteries at 100% SOC.

Cell	Onset Temp. (°C)	Thermal Runaway Temp. (°C)	Max. Temp. (°C)
1.2 Ah LCP	77	168	330
2 Ah LCO	82	185	485
2 Ah NCM	117	204	436
3 Ah LFP	106	219	455

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
