# Peer review of "High-Voltage Lithium-Ion Battery Using Substituted LiCoPO4: Electrochemical and Safety Performance of 1.2 Ah Pouch Cell"

_materials, 2020, doi:10.3390/ma13194450_

Round 1

Reviewer 1 Report

This manuscript reported the electrochemical and safety performance of 1.2 Ah pouch cell. A lot of electrochemical characterization methods of the resulting materials were made, also the electrochemical and thermal stability were studied. However, there are still some places that are not clear and some comments are made below for the authors to further enhance their paper before its possible publication.

  1. The subject of this work did not come-up properly in the introduction section. Therefore, introduction needs bit modification.
  2. The novelty must be supplemented, clearly, in the abstract.
  3. The reference [10] is used many times in the text, it will be better to explain that this work is continuation of your last work. Some information, particularly, for the experimental part are necessary to add it in this work. For example the section 2.1 authors must add all information of cell fabrication and electrochemical tests and then the reference, but mentioning just ‘’as described in ref[10]’’ is not appreciated for the reader.
  4. There are no coherence between the obtained resulted on TGA, DSC and XRD, also the scientific interpretation are weak, particularly for the XRD measurements, so a revision of this part is necessary to show and confirm between the obtained results, characterization and conclusion. It will be interesting to propose a mechanism or some scheme, explaining the results and performance of 1.2 Ah pouch cell. If not, the work risk being overlapped.
  5. In this paper, 1.2 Ah pouch cell showed high-performance on safety, stability and thermal stability. Please compare your results with those reported in the above listed literature.
  6. Finally, the conclusion must be logical outcome of results and discussion, showing the novelty of this work.

Reviewer 2 Report

The authors presented the fabrication of LiCoPO4-based high-voltage lithium-ion battery in the format of a 1.2Ah pouch cell that exhibited a highly stable cycle life at a cut-off voltage of 4.9 V. The high-voltage stability was achieved using a Fe-Cr-Si multi-ion-substituted LiCoPO4 cathode and lithium bis(fluorosulfonyl)imide in 1-methyl-1-propylpyrrolidinium bis(fluorosulfony)imide as the electrolyte.

It was evaluated the safety performance of the pouch cell under electrical abuse conditions and determined the thermal stability of delithiated LCP by accelerating rate calorimetry (ARC), thermogravimetric analysis (TGA), differential scanning calorimetry (DSC), and in situ X-ray diffraction (XRD) analyses

The manuscript contains descriptions of studies that have been planned, carried out and interpreted. There are sufficient details given to replicate the proposed experimental procedures.

I recommend that the manuscript is published.

Reviewer 3 Report

Row 38 Written as “…has been increased to 105 order [7]”. It seems to be correct to write down how …has been increased to 10^{5} order [7].

Row 77 Written as “…the frequency range 10 kHz to 50 mHz with…”. It seems to be correct to write down how … “…the frequency range 10 kHz to 50 MHz with…”

Round 2

Reviewer 1 Report

The authors have answered all my questions, revised paper can be accepted.